# Exploring the Fibrin(ogen)olytic, Anticoagulant, and Antithrombotic Activities of Natural Cysteine Protease (Ficin) with the κ-Carrageenan-Induced Rat Tail Thrombosis Model

**DOI:** 10.3390/nu14173552

**Published:** 2022-08-29

**Authors:** Hye Ryeon Yang, Du Hyeon Hwang, Ramachandran Loganathan Mohan Prakash, Jong-Hyun Kim, Il-Hwa Hong, Suk Kim, Euikyung Kim, Changkeun Kang

**Affiliations:** 1Department of Basic Veterinary Medicine, College of Veterinary Medicine, Gyeongsang National University, Jinju 52828, Korea; 2Institute of Animal Medicine, Gyeongsang National University, Jinju 52828, Korea

**Keywords:** ficin, natural cysteine protease, fibrinogen, thrombosis, κ-carrageenan

## Abstract

Although fibrinolytic enzymes and thrombolytic agents help in cardiovascular disease treatment, those currently available have several side effects. This warrants the search for safer alternatives. Several natural cysteine protease preparations are used in traditional medicine to improve platelet aggregation and thrombosis-related diseases. Hence, this study aimed to investigate the effect of ficin, a natural cysteine protease, on fibrin(ogen) and blood coagulation. The optimal pH (pH 7) and temperature (37 °C) for proteolytic activity were determined using the azocasein method. Fibrinogen action and fibrinolytic activity were measured both electrophoretically and by the fibrin plate assay. The effect of ficin on blood coagulation was studied by conventional coagulation tests: prothrombin time (PT), activated partial thromboplastin time (aPTT), blood clot lysis assay, and the κ-carrageenan thrombosis model. The Aα, Bβ, and γ bands of fibrinogen are readily cleaved by ficin, and we also observed a significant increase in PT and aPTT. Further, the mean length of the infarcted regions in the tails of Sprague–Dawley rats was shorter in rats administered 10 U/mL of ficin than in control rats. These findings suggest that natural cysteine protease, ficin contains novel fibrin and fibrinogenolytic enzymes and can be used for preventing and/or treating thrombosis-associated cardiovascular disorders.

## 1. Introduction

The genus *Ficus* consists of over 800 species and is one of approximately 40 genera of pantropic and subtropical origins [1]. Ficus fruits, roots, leaves, and lattices are used in traditional medicine to treat a variety of illnesses, including cancer, inflammatory, cardiovascular, and ulcerative diseases as well as gastrointestinal (colic, indigestion, loss of appetite, and diarrhea) and respiratory (sore throat, cough, and bronchial problems) conditions [2,3,4]. Ficin (EC 3.4.22.3) is a protease enzyme extract derived from fig latex (*Ficus carica*). It has many similarities to papain in terms of substrate selectivity, esterase activity, transpeptidase reactions, and activation by reducing agents [5,6].

Damage to blood vessels causes the plasma coagulation system and blood platelets to activate, which causes a blood clot made up of platelets and fibrin to form [7]. Numerous coagulation factors, such as fibrinogen and/or fibrin(ogen) and fibrin breakdown products, regulate leukocyte migration and cytokine production to control the inflammatory response [8]. While soluble fibrinogen is transformed into insoluble fibrin by thrombin during coagulation, fibrin deposits are eliminated by plasmin, which is primarily produced from soluble fibrinogen by the plasminogen activators during fibrinolysis [9]. Fibrinolysis plays a key role in the balance between coagulation and inflammation. As a result of tissue components being exposed in the lumen, ongoing inflammatory and remodeling processes eventually weaken the artery wall and lead to thrombotic events. The currently used fibrinolytic enzymes and thrombolytic agents in clinical settings have several side effects, including bleeding complications and hemorrhage [10]. This has led many researchers to search for safer alternatives from natural sources. Several natural cysteine protease preparations in traditional medicine have been used to improve platelet aggregation and thrombosis-related diseases. However, although natural cysteine proteases may have a therapeutic benefit for cardiovascular illnesses, there have been few to no significant human and animal trials to investigate these effects [11,12]. Therefore, the purpose of this study is to examine the beneficial effects of one such natural cysteine protease, ficin, using various experiments.

## 2. Materials and Methods

### 2.1. Chemicals and Reagents

Ficin, fibrinogen (type I-S from bovine plasma), thrombin (from bovine plasma), κ-carrageenan, plasmin, and streptokinase were purchased from Sigma-Aldrich (St. Louis, MO, USA). PT and aPTT combination test kits were obtained from Vetscan^®^VSpro (Abaxis Inc., Union City, CA, USA). All reagents used were of the purest grade.

### 2.2. Sodium Dodecyl Sulfate-Polyacrylamide Gel Electrophoresis (SDS-PAGE)

Electrophoresis was performed in accordance with the Laemmli protocol [13] using 12% separating gel and 4% stacking gel. Ficin was prepared in non-reducing sample buffer (4% SDS, 125 mM Tris-HCl, pH 6.8, 20% glycerol, and 0.01% bromophenol blue) and kept at −20 °C until use. Ficin was electrophoresed using the Tris-glycine running buffer for 90 min at 100 V. For molecular weight estimation, the molecular weight markers, 10–200 kDa (Precision Plus ProteinTM Standards, Bio-Rad, Hercules, CA, USA), were run in parallel with ficin. The gel was stained with 0.125% Coomassie blue in 40% methanol and 10% acetic acid after electrophoresis.

### 2.3. Proteolytic Activity Assay

Proteolytic activity was evaluated as described by Segers et al. with some modifications [14]. The reaction mixture contained 100 µL of ficin, 150 µL of reaction sodium bicarbonate buffer (0.5% solution, pH 8.3) (Buffer A), and 250 µL of 2.5% azocasein (*w*/*v*) dissolved in Buffer A. The assays were run at 37 °C, and after 30 min, they were terminated by adding 400 µL of 10% (*w*/*v*) trichloroacetic acid. The precipitated protein was removed by centrifuging the reaction mixture at 12,000 rpm for 20 min. The supernatant (500 µL) was neutralized by adding 300 µL of 500 mM sodium hydroxide, and the absorbance at 440 nm was measured using a UV/visible spectrophotometer (PowerWaveTMXS, BioTek Instruments, Inc., Winooski, VT, USA).

### 2.4. Effect of Temperature and pH on Protease Activity and Stability

In order to determine the optimum temperature and pH for ficin’s protease activity, the procedure described by Aissaoui, N. et al. was slightly modified [15]. Thermal stability was tested by incubating the ficin at temperatures ranging from 4 to 80 °C for 60 min. The pH stability of the ficin was tested by storing it at 4 °C for 30 min in several buffers with pH values ranging from 4.0 to 11.0 (pH 4, acetate buffer; pH 7, phosphate buffer; pH 10.0 and 11.0, glycine-NaOH buffer). Estimated residual proteolytic activity was represented as a fraction of the original activity, which was considered to be 100%.

### 2.5. Fibrin Plate Assay

Fibrinolytic activity was evaluated according to the method described by Astrup and Mullertz [16], with minor modifications. Briefly, plasminogen-free plates made by combining 5 mL of 0.6% *w*/*v* fibrinogen (Calbiochem, Darmstadt, Germany) in 50 mM Tris–HCl buffer (pH 7.8) with 1% agarose and 100 µL thrombin (100 NIH U/mL, Sigma-Aldrich). For the purpose of forming a fibrin clot, the plates were kept at room temperature (RT, 25 °C) for 30 min. Each of 10 μL ficin sample (0, 0.0125, 0.025, 0.05, 0.1 U) or positive control (plasmin 1 mg/mL) was dropped onto dry filter paper disks (diameter, 5 mm), which were then put over the fibrin layer and incubated for 24 h at 37 °C. Clear transparent zones suggested fibrin degradation, and the potency of fibrin degradation was proportional to its diameter. The diameters of the transparent rings were measured using the ImageJ software, and the fibrinolysis area (mm^2^) was calculated. The amount of lysis produced by each sample was estimated, and the mean diameter of the hydrolyzed clear zone was measured.

### 2.6. Examination of Fibrinolytic Activity Using Fibrin Zymography

Fibrin zymography was performed according to the Kim, Seung-Ho method [17]. Briefly, fibrinogen (0.6 mg/mL) and thrombin (0.01 unit/mL) dissolved in 20 mM sodium phosphate buffer (pH 7.4) was copolymerized with 12% polyacrylamide to prepare the respective zymography gel. The ficin was prepared for analysis using a non-reducing sample buffer, and then gels were run at 100 V at 4 °C. SDS was then removed from the gel after electrophoresis by washing it twice in 2.5% Triton X-100 for 30 min. Then, the gel was stained with 0.125% Coomassie blue after being treated with 20 mM Tris (pH 7.4), 0.5 mM calcium chloride, and 200 mM sodium chloride at 37 °C for 16 hr. The fibrinolytic areas were identified by the clear gel zones.

### 2.7. Fibrinogenolytic Activity

According to the method described by Matsubara et al., the fibrinogenolytic activity of ficin was examined [18]. In brief, 10 μL of ficin and 5 μL of reaction buffer were added to 15 μL of bovine fibrinogen (20 mg/mL) and incubated at 37 °C for the designated period of time and doses (pH 7.4, 200 mM sodium chloride, 0.5 mM calcium chloride, and 20 mM Tris). The digested products were examined using a 7.5% SDS-PAGE.

### 2.8. Blood Clot Lysis Assay

Blood clot lysis assay was used with a slightly modified method of Prasad, Sweta, et al. [19]. Healthy dogs’ venous blood was taken and placed in several pre-weighed, sterile microcentrifuge tubes (500 μL/tube), where it was then incubated for 45 min at 37 °C. Each tube containing a blood clot was weighed again to calculate the clot weight (clot weight = weight of clot-containing tube—weight of tube alone) after the serum had been fully removed. The clot was placed into each appropriately labeled microcentrifuge tube, and different concentrations of ficin were added to the tubes (0–0.1 U). Each tube containing clots had streptokinase and phosphate-buffered saline added as negative and positive controls, respectively. For 24 h, all of the tubes were incubated at 37 °C while being watched for clot lysis. After the fluids acquired during incubation were taken out, the tubes were once more weighed to determine the weight difference following clot breakup. The percentage of clot lysis was calculated using the weight difference between before and after clot lysis.

### 2.9. PT/aPTT Measurement

Prothrombin time (PT) and the activated partial thromboplastin time (aPTT) were measured to explore how ficin affected blood coagulation. Blood samples from healthy dogs were collected for the coagulation test in a solution of 3.2% sodium citrate (1:9) mixed with ficin at room temperature for 10 min. After incubation, a PT and aPTT combination test kit was used to examine the mixtures (Vetscan^®^VSpro PT/aPTT combination test cartridge, Abaxis Inc., Union City, CA, USA).

### 2.10. Experimental Animals

Four-week-old male Sprague–Dawley rats were obtained from Samtako Inc. (Osan, Korea) and housed in the lab of the Gyeongsang National University animal research facility. Rats were kept in cages with a conventional chow meal (8% protein, 5% fat, 4.5% fiber, 8% carbohydrate, 0.7% calcium, 1.2% phosphorus, and 14% moisture), water accessible at all times, and a 12 h light/dark cycle. The ambient temperature was 23 ± 2 °C, and the relative humidity ranged from 35 to 60%. The animal study protocol used in this work was approved by the Institutional Animal Care and Use Committee of Gyeongsang National University, and the protocol number is GNU-200122-R0003.

### 2.11. κ-Carrageenan-Induced Rat Tail Thrombosis Model

A total of 16 SD rats were randomly allocated into four groups (*n* = 4 per group): group 1, vehicle-treated group (Control); groups 2 and 3, 1 U/kg and 10 U/kg ficin with κ-carrageenan, respectively; group 4, streptokinase with κ-carrageenan group (positive control). The rat dorsal tail vein received an injection of κ-carrageenan that had been dissolved in saline. Following the ligation of a location 12 cm from the tip of the rat tail, 1 mg/kg of κ-carrageenan was administered intravenously in order to measure thrombogenesis. After 20 min, the ligation was removed. The experimental group was pretreated with ficin for 1 h, followed by κ-carrageenan injection. A dose of 2000 U (mL/kg) of streptokinase was administered intraperitoneally, 1 mg/kg of κ-carrageenan was administered intravenously to the positive control group, and the rats’ tails were tied for 10 min. The incidence of infarctions and the size of the infarcted zone at the tail tip were assessed 48 h after the injection of κ-carrageenan.

### 2.12. Statistical Analysis

The results are expressed as the mean ± standard deviation (S.D.). A paired Student’s *t* test was used to assess the significance of the differences between two mean values. * *p* < 0.05 and ** *p* < 0.01 were considered statistically significant.

## 3. Results

### 3.1. SDS-PAGE Profile of Ficin

Figure 1A shows the SDS-PAGE gel electrophoretograms obtained under reducing conditions. Ficin appears to have a molecular weight of around 24 kDa. To investigate the fibrinolytic activity of ficin, fibrin zymography was performed using various concentrations of ficin. The fibrinolytic activity showed that ficin was highly effective in degrading fibrin, and its molecular weight was over 50 kDa (Figure 1B). These results suggest that ficin can degrade fibrin and thus could act as a fibrinolytic enzyme.

### 3.2. Effect of Temperature and pH on Protease Activity and Stability

We investigated the stability of ficin at various temperatures and pH levels using the azocasein test. As indicated in Figure 2A, the ficin was carried out by incubating the enzyme for 60 min at a temperature range of 4 to 80 °C. Ficin was stable up to 37 °C and then decreased slightly at 60 °C. Incubation of ficin for 60 min at a higher temperature (80 °C) resulted in a significant decrease in its activity.

It was tested in several buffer systems with varying pH values to establish the ideal pH for ficin action (see Materials and Methods). A pH of 7 was optimum for ficin using a phosphate buffer (Figure 2B). The activity steadily decreased at pH 10.0. Thus, the stability of ficin changed with the increase in pH and temperature, with a maximum at pH 7 and 60 °C, respectively.

### 3.3. Fibrinolytic Activity of Ficin

To determine whether the ficin had direct or indirect fibrinolytic activity, it was allowed to react on fibrin plates (Figure 3). A clear zone was formed by ficin in a dose-dependent manner, and it had more fibrinolytic activity than plasmin (positive control). The fibrinolytic activity of ficin was found after 24 h of incubation, and the formation of a clear hollow was gradually observed from 24 h onwards. Thus, it was confirmed that ficin has fibrinolytic activity.

### 3.4. Fibrinogenolytic Activity of Ficin

Ficin was incubated with human fibrinogen at 37 °C to investigate its ability to hydrolyze it, and the reactant was run on a 7.5% SDS-PAGE. Ficin degraded α-and β-chains of fibrinogen in a dose-dependent manner (Figure 4). The α and β chain of fibrinogen was immediately digested in 0.0125 U of ficin. The complete degradation of the γ-chain of fibrinogen was observed at 0.05 U after incubation with ficin.

### 3.5. Blood Clot Lysis

To evaluate the effects of ficin on blood clots, we performed a dog blood clot lysis assay. As shown in Figure 5, ficin demonstrated dose-dependent clot lysis activity. The clot lysis ability of ficin was significantly higher than that of the PBS-treated group (negative control) and similar to that of the streptokinase-treated group (positive control). Ficin significantly decreased the weight of the clot, especially 0.1 U/mL of ficin, which dissolved 98.6% of the blood clot.

### 3.6. Anti-Coagulation Effect of Ficin

To examine the effect of ficin on anticoagulant parameters, aPTT and PT levels were measured using dog blood. Treatment with ficin delayed the PT and aPTT values compared to those not treated with ficin (Table 1). The indicated concentration of ficin (0.8 U/mL) prolonged the PT and aPTT above 35 s and 200 s, respectively. Thus, ficin has components acting as anticoagulants through the regulation of the intrinsic and extrinsic coagulant pathways.

### 3.7. κ-Carrageenan-Induced Rat Tail Thrombosis Assay

The antithrombotic effect of ficin was investigated using a model of κ-carrageenan-induced rat tail thrombosis. After being treated with κ-carrageenan, the rats’ tails took on an auburn color, indicating thrombus development (Figure 6). The negative control group (κ-carrageenan administered alone) had an average thrombus length of 11.8 ± 0.3 cm. In this group, thrombus in the tail tip changed color from wine to auburn and progressed to significant necrosis 48 h after the injection of κ-carrageenan. However, ficin efficiently attenuated these symptoms. In the ficin-treated group (10 U/kg), the mean thrombus length was 8.8 ± 0.3 cm. The mean thrombus length in the streptokinase-treated group (positive control) was 10.0 ± 0.5 cm. Thus, ficin showed a potent antithrombotic effect in a κ-carrageenan-induced rat tail thrombosis model.

## 4. Discussion

Natural cysteine proteases, such as ficin, are required for latex coagulation following biotic or abiotic injuries [20]. Because of their numerous physiological functions and uses in a variety of industrial sectors such as the pharmaceutical industry, proteolytic enzymes have recently attracted more attention [2,21]. Beyond their potential use in clinical laboratories, research using these proteolytic enzymes may help generate compounds useful for treating coagulation-related illnesses [22].

Fibrin, the major ingredient in blood clots, is broken down by fibrinolytic enzymes. Myocardial infarction and other heart illnesses, which are the main causes of mortality globally, are caused by the thrombosis that comes from the accumulation of fibrin in blood vessels [16,23]. Successful isolation of fibrinolytic enzymes from numerous sources has been obtained [24]. Streptokinase, urokinase, pro-urokinase, reteplase, and alteplase are examples of fibrinolytic enzymes and thrombolytic drugs currently used in clinical settings. These drugs have significant unintended physiological side effects, including excessive bleeding, a short plasma half-life, limited fibrin specificity, and high therapeutic doses. [25,26]. Therefore, the need to discover less expensive, safer fibrinolytic enzymes and thrombolytic medicines derived from natural sources is important.

Several reports have demonstrated the use of ficin in traditional medicine. Hence, in this study, we explored the fibrinolytic and thrombolytic potential of ficin. The findings of the current research suggested that ficin is a possible source of fibrinolytic and thrombolytic agents. Both in the fibrin plate assay and thrombolytic assay with whole blood, ficin showed similar and close results with respect to that of the standard drugs (plasmin and streptokinase). When the whole blood clot was incubated with ficin (0.0125–0.1 U) against the streptokinase (1000 U/mL), it was able to dissolve 99% of the whole blood clot. Fibrin zymography showed that ficin degrades fibrin, thus revealing fibrinolytic activity with clear bands. Ficin also showed notable fibrinogenolytic property: the Aα and Bβ subunits were susceptible, whereas the γ subunits were completely hydrolyzed by increasing the dose.

PT and aPTT are hemostatic indices that provide insight into the coagulation status of individuals [27], and these tests are usually used to screen for coagulation factor deficiencies [28,29]. We examined at how ficin affected the internal and external coagulation pathways. The coagulation factors FII, V, VII, and X are the primary extrinsic coagulation factors that are reflected in PT. The intrinsic coagulation system, which involves factors VIII, IX, XI, and XII, functions as a screening test for aPTT. Ficin (0.8 U/mL) possesses anticoagulant effects, which are demonstrated by the prolonging of the PT and aPTT. This indicates that it suppressed the clotting process by acting on the pathway of both exogenous and endogenous coagulation systems while blocking the conversion of fibrinogen to fibrin. Thus, ficin could have an anticoagulant effect and acts on endogenous anticoagulation.

When κ-carrageenan is administered intravenously, it triggers immune cells to produce histamine and serotonin, which causes acute blood vessel inflammation in a variety of animals [30,31,32]. The rat tail also showed signs of dry necrosis and a wine-colored zone near the tip after intravenously injecting κ-carrageenan. We used the κ-carrageenan-induced rat tail thrombus model to assess the antithrombotic activity of ficin in vivo. In rat tails, ficin has an antithrombotic effect that is dose-dependent and either delays the formation of thrombi or dissolves those that are already present. Thus, ficin is, in fact, an effective antithrombotic reagent with great exploitable potential.

In conclusion, the results presented here show that natural cysteine proteases, such as ficin from *Ficus carica* latex, have different fibrin(ogen)olytic activities as well as anticoagulant and antithrombotic effects. This preliminary investigation on ficin’s impact on the coagulation and fibrinolysis systems has a wide range of implications. Chiefly, it reveals novel plant species as possible sources of pharmacological treatments for a variety of hemostatic illnesses because of their combined roles in wound healing and hemostasis. Furthermore, ficin is attractive owing to its antithrombotic effect in a κ-carrageenan-induced rat tail thrombus model, and it could be a considerable candidate for the development of new antithrombotic medicine. Further research is needed to look at the effects of ficin on the thrombin time test and platelet aggregation tests and to investigate its mechanism of action at the molecular level.

## Figures and Tables

**Figure 1 nutrients-14-03552-f001:**
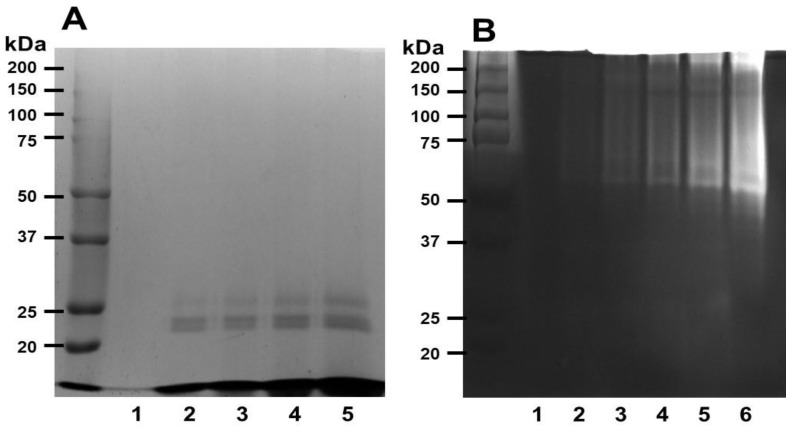
SDS-PAGE profile and fibrinolytic activity of ficin. (**A**) SDS gel electrophoresis of ficin under reducing condition. The gels were stained with 0.125% Coomassie blue. (**B**) Fibrin zymography with various concentrations of ficin. Clear zones in the fibrin gel indicate regions of proteolytic activity. Lane 1—0, lane 2—0.0125, lane 3—0.025, lane 4—0.05, lane 5—0.1, lane 6—0.2 (U/mL).

**Figure 2 nutrients-14-03552-f002:**
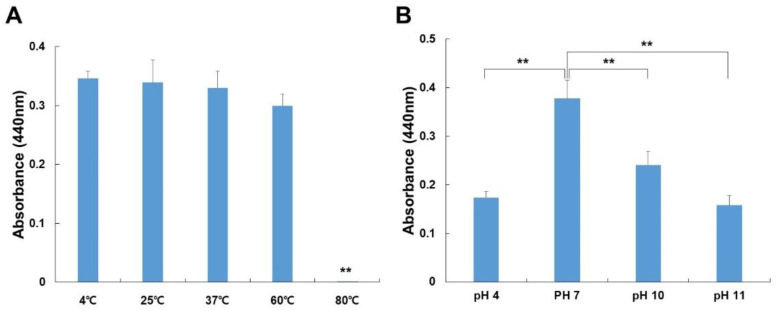
Effects of temperature (**A**) and pH (**B**) on the enzyme activity of ficin. The enzyme activity was measured by azocasein assays at 440 nm. (**A**) The ficin activity was assessed after incubation at temperatures ranging from 4 to 80 °C. (**B**) The ficin activity was analyzed by incubating at 37 °C for 30 min over a pH range of 4, 7, 10, and 11. The data shown are the mean ± S.D. of three independent experiments. The asterisk indicates the presence and levels of significant differences (** *p* < 0.01) from other groups.

**Figure 3 nutrients-14-03552-f003:**
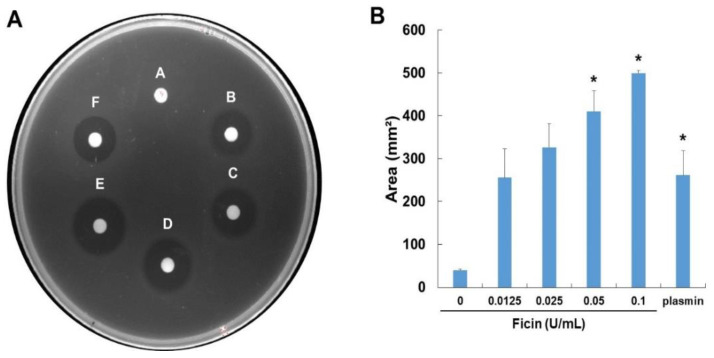
Fibrinolytic activity of ficin. (**A**) Fibrinolytic activity was assayed on the fibrin plate. Samples were applied to the disc in the plate and allowed to incubate at 37 °C for 24 h. (**B**) Fibrinolytic activity was measured by the dimension of clear zones. A—0, B—0.0125, C—0.025, D—0.05, E—0.1 (U), F—plasmin 2 U/mL. The data shown are the mean ± S.D. of three independent experiments. * *p* < 0.05 is considered a significant difference from the control group.

**Figure 4 nutrients-14-03552-f004:**
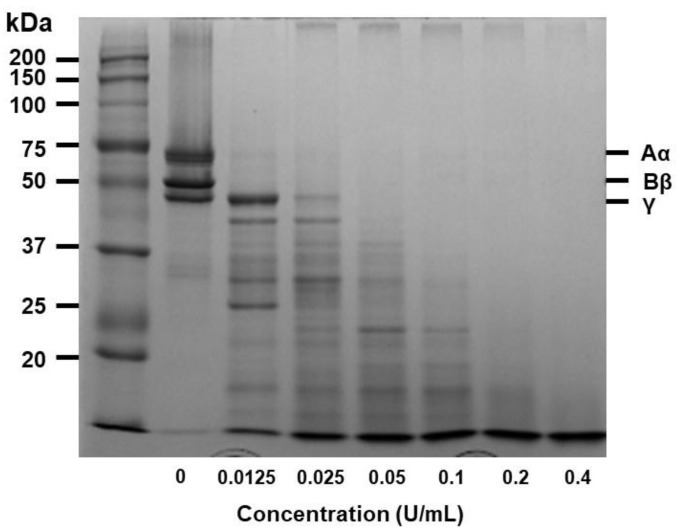
Dose-dependent effect of ficin on fibrinogen. The fibrinogenolytic activity was evaluated on an SDS-PAGE after incubation of various concentrations of ficin with bovine fibrinogen at 37 °C for 30 min. The mixed sample electrophoresis in 7.5% SDS-PAGE and Coomassie blue staining. Fibrinogen consists of three polypeptides: chains Aα, Bβ, and γ.

**Figure 5 nutrients-14-03552-f005:**
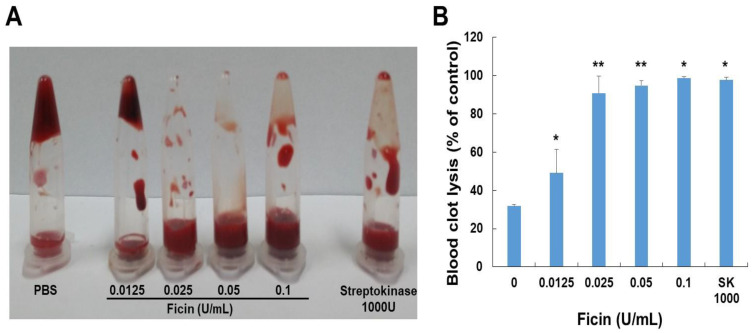
(**A**) Effect of ficin in in vitro thrombolysis. (**B**) Clot lysis by various concentrations of ficin and streptokinase (positive control) observed for 24 h. The data shown are the mean ± S.D. of three independent experiments. * *p* < 0.05 and ** *p* < 0.01 are considered a significant difference from the control group.

**Figure 6 nutrients-14-03552-f006:**
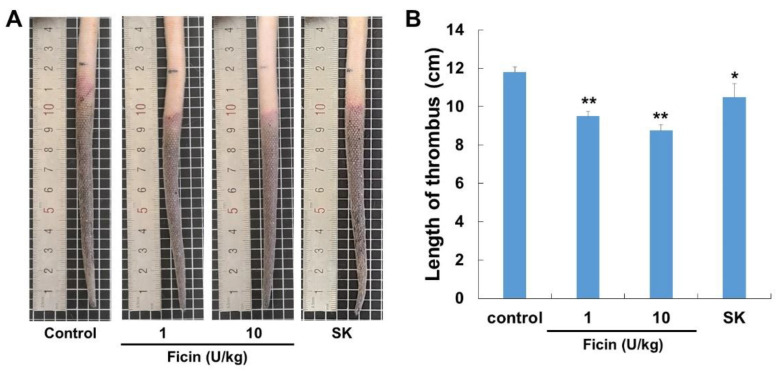
Ficin inhibits κ-carrageenan-induced thrombosis in Sprague–Dawley rats. (**A**) Representative photographs 48 h after carrageenan injection. (**B**) Bar diagram showing the inhibitory effect of ficin and streptokinase in tail thrombus at 48 h. Data are expressed as the mean ± S.D. (*n* = 4), * *p* < 0.05 and ** *p* < 0.01 are considered a significant difference from the control group.

**Table 1 nutrients-14-03552-t001:** Anticoagulation test using dog blood. aPTT and PT of dog blood incubated with ficin at the indicated concentrations. Fresh, citrated dog blood and indicated concentration of ficin (0.8 U/mL) was preincubated for 10 min. The data shown are the mean ± S.D. of three independent experiments. s = Seconds.

Ficin	PT (s)	aPTT (s)
Control	17.5 ± 0.8	93.9 ± 2.1
0.8 U/mL	35<	200<
Normal range	14–19	75–105

## Data Availability

Not applicable.

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
