# Peer review of "Exploring the Fibrin(ogen)olytic, Anticoagulant, and Antithrombotic Activities of Natural Cysteine Protease (Ficin) with the κ-Carrageenan-Induced Rat Tail Thrombosis Model"

_nutrients, 2022, doi:10.3390/nu14173552_

Round 1

Reviewer 1 Report

Reviewers report

Title: Exploring the Fibrin(ogen)olytic, Anticoagulant and Antithrombotic Activities of Natural Cysteine Protease (Ficin) With the κ-Carrageenan-Induced Rat Tail Thrombosis Model

Introduction:

This section is well written and with clearly stated objectives.  

Materials and methods:

Section 2.1: Please completely write all chemicals and reagents used in this study.

Section 2.4: Please provide a proper reference and explain on which basis you chose this temperature range (10-80C)

Section 2.6: Please provide a proper reference.

Section 2.8: Please provide a proper reference.

Section 2.10: Please provide the number of rodents and also provide the compositional information of diets (supplementary information) which was fed to these rats. 

Result:

The length of results section is unexpectedly large especially when you also have comprehensive discussion section. I suggest authors to reduce the length of results at least by ¼.

-Since the results in this section varied at different significance levels authors should include the p-values in their representation of results in all comparisons.

Discussion:

-From line 255 to 272, I suggest authors to shorten it to few lines and authors should directly jump to elaborating their own results only.

-Please shorten your results section and elaborate your experiments results more comprehensively in discussion section.   

Author Response

Response to reviewers 1:

To reviewer 1: Thank you very much for your valuable comments. According to your comments, we have changed the manuscript as follows. In the manuscript, revised portions are shown in red color.

Point 1. Section 2.1: Please completely write all chemicals and reagents used in this study.

-Answer: Thank you for the valuable comment. We have added all the used chemicals and reagents in this section.

Point 2. Section 2.4: Please provide a proper reference and explain on which basis you chose this temperature range (10-80C)

-Answer: Thank you for the valuable comment. We have added a proper reference and chosen this temperature range (4, 25, 37, 60 and 80 ℃) according to the reference.

Point 3. Section 2.6: Please provide a proper reference.

-Answer: Thank you for the valuable comment. We have added a proper reference as you suggested.

Point 4. Section 2.8: Please provide a proper reference.

-Answer: Thank you for the valuable comment. We have added a proper reference as you suggested.

Point 5. Section 2.10: Please provide the number of rodents and also provide the compositional information of diets (supplementary information) which was fed to these rats. 

-Answer: Thank you for the valuable comment. We used a total 16 SD rats and divided them into 4 groups with four rats each. And each group of rats was fed normal rat chow diet (8% protein, 5% fat, 4.5% fiber, 8% carbohydrate, 0.7% calcium, 1.2% phosphorus, and 14% moisture) during the experiment period. We also added the sentences in sections 2.10 and 2.11.

Point 6. The length of results section is unexpectedly large especially when you also have comprehensive discussion section. I suggest authors to reduce the length of results at least by ¼.

-Answer: Thank you for the valuable comment. We have corrected and reduced the sentences as you suggested.

Point 7. Since the results in this section varied at different significance levels authors should include the p-values in their representation of results in all comparisons.

-Answer: Thank you for the valuable comment. We have rechecked the significance levels and included the p-value in each figure legend.

Point 8. From line 255 to 272, I suggest authors to shorten it to few lines and authors should directly jump to elaborating their own results only.

-Answer: Thank you for the valuable comment. We have deleted several lines to match the context of a sentence in English.

Point 8. Please shorten your results section and elaborate your experiments results more comprehensively in discussion section.  

-Answer: Thank you for the comment. We have revised those sentences as you suggested. 

Reviewer 2 Report

In my opinion the article is well written and the topic is well developed

Author Response

Response to reviewers 2:

To reviewer 2: Thank you very much for your valuable comments. We corrected several typos and errors in our resubmitted manuscript. In the manuscript, revised portions are shown in red.